# Bleeding and Thrombosis in Multiple Myeloma: Platelets as Key Players during Cell Interactions and Potential Use as Drug Delivery Systems

**DOI:** 10.3390/ijms242115855

**Published:** 2023-11-01

**Authors:** Anushka Kulkarni, Despina Bazou, Maria José Santos-Martinez

**Affiliations:** 1The School of Pharmacy and Pharmaceutical Sciences, Trinity College Dublin, The University of Dublin, D02 PN40 Dublin, Ireland; kulkaran@tcd.ie; 2School of Medicine, University College Dublin, D04 V1W8 Dublin, Ireland; despina.bazou@ucd.ie; 3School of Medicine, Trinity College Dublin, D02 R590 Dublin, Ireland; 4Trinity Biomedical Sciences Institute, Trinity College Dublin, D02 R590 Dublin, Ireland

**Keywords:** platelets, multiple myeloma, cancer, drug carrier

## Abstract

Multiple myeloma (MM) is a hematological malignancy originated in the bone marrow and characterized by unhindered plasma cell proliferation that results in several clinical manifestations. Although the main role of blood platelets lies in hemostasis and thrombosis, platelets also play a pivotal role in a number of other pathological conditions. Platelets are the less-explored components from the tumor microenvironment in MM. Although some studies have recently revealed that MM cells have the ability to activate platelets even in the premalignant stage, this phenomenon has not been widely investigated in MM. Moreover, thrombocytopenia, along with bleeding, is commonly observed in those patients. In this review, we discuss the hemostatic disturbances observed in MM patients and the dynamic interaction between platelets and myeloma cells, along with present and future potential avenues for the use of platelets for diagnostic and therapeutic purposes.

## 1. Multiple Myeloma: Clinical Manifestations and Treatment 

Multiple myeloma (MM) is a bone marrow malignancy that accounts for 1% of all cancer types. An average of 80 thousand people are diagnosed with MM in Europe and the United States every year [1,2,3] and it is more common in people above 60 years of age, particularly in men [4]. 

The disease is characterized by the unhindered proliferation of B cells in the bone marrow. It can affect several parts of the body including, but not limited to, the spine, ribs, kidney, blood, and brain, leading to several clinical manifestations including hypercalcemia, renal failure, anemia, and bone pain due to the presence of lytic lesions (CRAB). The diagnostic criteria for active MM include the presence of more than 60% of clonal bone marrow plasma cells, a serum free light chain ratio (FLC) ≥ 100, and the presence of at least one focal lesion of 5 mm or greater identified by magnetic resonance imaging (MRI) [5]. 

MM usually begins as an asymptomatic premalignant condition known as monoclonal gammopathy of unknown significance (MGUS) and smoldering multiple myeloma (SMM). Patients with MGUS and SMM usually have excessive circulating quantities of monoclonal antibodies, and elevated serum and urine microglobulin levels without the presence of clinical symptoms [6,7]. SMM is associated with a higher level of infiltrating plasma cells (from 10% to less than 60%) and higher microglobulin levels in serum and urine. Progression of MGUS (less than 10% of infiltrating plasma cells) to symptomatic MM is low, 1% per year, while the percentage of patients that progress to MM among SMM patients is ten times higher [8]. The progression toward MM is associated with the presence of various molecular markers [9]. The International Staging System for MM takes into consideration factors such as the gradual acquisition of cytogenetic abnormalities (translocations, insertion/deletion of chromosomes), the number of plasma cells in the bone marrow (BMPC), serum and urine M- protein levels, serum FLC ratio, and β2 microglobulin and albumin levels as criteria for establishing the patient stage (Table 1). 

Various molecular markers have been shown to confer poor prognosis. These include the deletion of chromosome 13q, 17p [11,12], and the presence of increased levels of serum beta macroglobulin and LDH [13]. 

The International Myeloma Foundation and European Hematology Association (EHA) and the European Society for Medical Oncology (ESMO) have laid down guidelines for the management of MM patients, based on published data from clinical trials [14,15]. For newly diagnosed patients, triplet therapy, including an immunomodulatory drug (IMID), e.g., lenalidomide, a proteasome inhibitor (PI), e.g., bortezomib, and the corticosteroid dexamethasone, is preferred, followed by autologous stem cell transplantation. For second-line treatment, a combination of different IMIDs and PIs is implemented followed by PI-based regimens [16]. Immunotherapies with monoclonal antibodies like daratumumab, elotuzumab, and isatuximab are also included in the treatment plan. Recently, bispecifics have shown great responses in relapsed/refractory MM patients with one to three prior therapies. The US-FDA and European Commission conditionally approved Talquetamab, marketed by Janssen under the name Talvey^®^. Talquetamab is a bispecific antibody, the first in its class to target the novel receptor GPRC5D (NCT04634552) [17]. Similarly, both Janssen’s Teclistamab and Pfizer’s Elrexfio (elranatamab-bcmm) bispecific antibodies target the B-cell maturation antigen (BCMA)-cluster of differentiation three (CD3) (NCT04557098, NCT04649359) in relapsed/refractory MM patients [18,19].

## 2. Hemostatic Complications Associated with Multiple Myeloma

Cancer-associated thrombosis (CAT) is frequently observed in cancer patients. In fact, venous thromboembolism (VTE) is commonly associated with disease progression and increased mortality. CAT has a complex underlying mechanism that has been associated with the secretion of tissue factor (TF), increased platelet activation, and endothelial cell dysfunction, among others [20]. VTE is commonly presented in patients with MM, even at the initial stages of MGUS [21]. VTE risk is especially higher within the first year of MGUS and active disease diagnosis. However, VTE is not an individual risk factor for the progression of MGUS to active MM [22]. Risk-predicting models (IMPEDE [23], SAVED [24], and PRISM score [25]) can be used for predicting the risk of VTE development in MM patients. However, extensive validation of these models is required before clinical use. Kapur et al. [26] showed that none of the models accurately predicted the VTE risk of MM patients and many patients in the intermediate and low-risk groups still developed VTE.

While the mechanisms underlying thrombosis have been extensively explored in solid tumors, these are still under-studied in MM. Cancer cells in solid tumors contribute to thrombosis by secreting TF and cancer procoagulant (CP). TF is constitutively expressed in different cancers such as leukemia, breast, pancreatic, brain, prostate, and ovarian cancer, apart from being secreted by monocytes and endothelial cells [27,28,29,30,31]. TF has been associated with the development of VTE in ovarian cancer patients [30]; however, no such correlation was found in patients with brain tumors [31]. TF can also bind to and activate factor VII, thus activating the coagulation cascade [32]. CP is a serine protease found in some cancers such as acute myeloid leukemia and Lewis lung carcinoma, and can directly activate factor X, thus contributing to the hypercoagulable state of the disease [20,33,34]. In MM, increased levels of thrombin generation, vWF (which is also elevated in solid tumors), and fibrin polymerization have been employed as potential indicators of thrombosis. Van Marion et al. argued that these alone do not contribute to the development of VTE, rather, patient- and treatment-specific factors should be taken into account [35,36]. In addition, Ghansah et al. suggested that resistance to activated protein C (APC) can lead to a hypercoagulable MM state as APC did not attenuate thrombin generation in MM patients [37]. More recently, Nielson et al. argued that EVs from MM patients exert procoagulant activity by inducing thrombin generation and carrying TF and procoagulant phospholipids. The researchers also found that this activity diminished during the course of treatment with induction therapy consisting of bortezomib, cyclophosphamide, and dexamethasone [38].

On the other hand, bleeding is also occasionally seen in MM patients. When compared to individuals with solid tumor malignancies, patients with hematological cancers are more likely to experience thrombocytopenia. Patients with MM have been shown to have the highest rates among those with hematological malignancies [39,40]. In addition to thrombocytopenia, dysfibrinogenemia brought on by the interaction of numerous paraproteins with the proteins in the coagulation cascade can also contribute to bleeding in MM patients [41,42,43,44].

Interestingly, there is a recent study by Li et al. [45] that compared the thrombin generation profile in whole blood and poor platelet plasma of 21 MM patients with healthy controls to understand the paradoxical existence of both bleeding and thrombosis in MM. They used a new method to measure thrombin generation in whole blood that takes into account the potential involvement of platelets and red and white blood cells during this process. The authors concluded in their work that “patients balance on a thin line between pro and anti-coagulant phenotype” and hypothesize that both platelets (thrombocytopenia and platelet dysfunction) and red blood cells (that promote thrombin generation in vivo) play a crucial role in this balance.

## 3. Hemostatic Complications Associated with Multiple Myeloma Treatment

While the current therapeutic agents used for treating MM have consistently demonstrated a tolerable safety profile with significant and clinically relevant benefits, thrombosis and bleeding in MM patients can also be attributed to chemotherapy.

Treatment with immunomodulatory agents like lenalidomide and dexamethasone has increased the prevalence of VTE in MM. A two-to-four-fold increased platelet activity in lenalidomide-treated MM patients has been observed, as assessed by an increased aggregation response and P-selectin upregulation. A significant increase in coagulation activator markers when compared to controls (patients at diagnosis) was also reported [46,47,48].

Bortezomib has been shown to induce reversible thrombocytopenia, probably due to the inhibition of platelet formation from megakaryocytes and an increase in the expression of GTPase Rho. [49] Although some patients on bortezomib may require platelet transfusion [50], concomitant therapy with high doses of dexamethasone has been reported to reduce the need for transfusions [51]. An in vitro study carried out on platelet-rich plasma from healthy human volunteers has also shown that bortezomib can inhibit ADP-induced platelet aggregation [52]. In three extremely rare cases, bortezomib has also been associated with severe diffuse alveolar hemorrhage in MM patients [53].

Daratumumab, an antibody that targets CD38 and induces direct antimyeloma activity, has been associated with a higher incidence of thrombocytopenia [54].

Guidelines recommend using aspirin or low-molecular-weight heparins (LMWHs) in MM patients [55]. Bleeding can also be associated with anticoagulant therapy. However, a study carried out on 1605 patients on warfarin (45.7%), LMWHs (29%), and direct orally acting anticoagulants (25.3%) demonstrated that only a small percentage (3.9%) of patients suffered from bleeding during a follow-up period of 1.6 years [56].

## 4. Platelet–Myeloma Cell Interactions

Platelets play a dynamic role in the bone marrow microenvironment by interacting with other cells locally. Their interaction with hematopoietic stem cells can occur through the release of platelet-derived growth factor (PDGF), transforming growth factor β-1 (TGF-β1), and vascular endothelial growth factor (VEGF), all of them involved in the cell growth and survival of hematopoietic cells [57]. Platelets and leukocytes also interact with each other by physical contact and via the release of chemical mediators [58,59]. For instance, the binding of platelets to monocytes via P-selectin (CD62P) leads to the release of the platelet-activating factor and TF, resulting in the activation of more platelets [60].

There are limited studies investigating the interaction between platelets and myeloma cells. In 2018, Takagi et al. showed that several MM cell lines (MM.1S, KMS-11, U266, OPM-2, and H929) were able to induce platelet aggregation [61]. In addition, when these cell lines were co-cultured with platelets or platelet releasate from healthy donors, cell proliferation was significantly enhanced, except for MM.1S. The authors then injected OPM-2 cells pre-exposed to platelets into mice and observed a high rate of tumor cell engraftment that resulted in decreased mice survival. They attributed this effect to IL-1β upregulation as the IL-1β OPM-2 knockout cells abrogated tumor engraftment [61].

MM cells release TNF-α [62] which is a potent platelet activator [63]. There are, however, contradictory results concerning platelet activation in the premalignant MGUS state and active MM disease. In the premalignant MGUS stage, the upregulation of P-selectin on the platelet surface and the abundance of plasma-soluble P-selectin levels have been reported [48,64,65]. On the other hand, platelets from patients with active MM have shown impaired platelet activation as evaluated by P-selectin expression in response to several agonists including collagen, ADP, epinephrine, and protease-activated receptor-1 (PAR-1)-activating peptide [66]. In another study, there was no significant difference in P-selectin expression between healthy donors and MM patients at diagnosis [46]. Myeloma cells can also produce IL-6 [67]. Apart from inhibiting apoptosis in MM cells [68] and increasing RANKL [69], IL-6 can increase the platelet count by increasing thrombopoietin production [70]. It is therefore imperative that further studies are required to elucidate the role of platelets in the different MM disease stages.

Given the limited number of studies exploring the molecular interactions between MM cells and platelets, we propose that this interaction could take place through several mechanisms, as depicted in Figure 1. MM cells express platelet-derived growth factor receptor (PDGFR) α and β which could bind PDGF released by activated platelets [71]. The upregulation of this receptor is also strongly correlated to angiogenesis and microvessel density (MVD) [72]. P-selectin glycoprotein ligand-1 (PSGL-1) expressed on MM cells could bind P-selectin on activated platelets. Leukocytes and myeloid cells also express PSGL-1 that could bind to activated platelets [73]. It has been reported that MM patients with no PSGL-1 expression had a lower survival; however, a potential mechanism behind this was not reported [74]. Syndecan-1 (CD138) is also expressed on the surface of MM cells and when shed, it promotes tumor growth and metastasis. Syndecan-1 can also bind to the angiogenic factors, VEGF, and fibroblast growth factor-2 (FGF-2), promoting angiogenesis and negatively affecting patient prognosis [75,76].

Platelet factor-4 (PF4) is stored in the platelet alpha granules and released upon activation. In MM patient serum samples, a downregulation of PF4 has been reported [77]. However, human recombinant PF4 has shown proapoptotic activity by inhibiting MM cell proliferation and angiogenesis, through the inhibition of the STAT3 and IL6-STAT3 pathways, both in vitro and in vivo [78]. Platelets also release sphingosine-1 phosphate (S1P) that acts as a tumorigenic and angiogenic growth factor. In myeloma, S1P binds to SIP1, SIP2, and SIP3 receptors on myeloma plasma cells, preventing dexamethasone-induced apoptosis [79,80]

## 5. Platelets as Diagnostic and Therapeutic Tools in Cancer

Cells have been engineered to be used as drug delivery vehicles due to their extended stability, minimal immunogenicity, and selective targeting [81]. Utilizing platelets as drug delivery vehicles appears to be a promising alternative since they have been implicated in the proliferation and spread of tumor cells.

Platelet microvesicles (PMVs) are released by activated platelets. Microvesicles are extracellular vesicles (EVs) that range from 100 to 1000 nm and are released by the outward blebbing of the cell membrane. It is intriguing that despite the platelets’ absence of a nucleus, PMVs may transport genetic material like RNA, mRNA, and microRNA. PMVs can interact with tumor cells as well as circulating immune cells via receptors such as CXCR4, which aids in the growth and spread of tumors [82,83], and GPIIb/IIIa, which enhances tumor survival and migration [84]. PMVs also contain lipid mediators, such as eicosanoids, which may be utilized as biomarkers [85]. Furthermore, they are nearly 100 times more procoagulant than platelets due to their higher prothrombotic potential [86]. Due to their small size, they can enter tissues more readily. Recently, due to their excellent stability and low immunogenicity, platelet EVs (PEVs) have demonstrated the potential to be employed as drug carriers. Both hydrophilic and hydrophobic drugs have been successfully loaded into PEVs, mainly through two methods: (a) platelet incubation with drugs and (b) drug loading directly into EVs through sonication, freeze–thaw cycles, dialysis, and electroporation. [86,87].

In addition to PMVs [88], platelets themselves can also be employed as drug carriers because they interact with tumor cells via tumor cell-induced platelet aggregation (TCIPA), promoting the passive delivery of the drug to the tumor site [89]. This approach has been successful in preliminary studies where platelets have been loaded with doxorubicin and subsequently delivered to lymphoma cells in vitro [90]. In this study, doxorubicin was loaded through the open canalicular system with a high loading and encapsulation efficiency. In vivo studies using athymic BALB/c-nude mice have demonstrated the ability of the doxorubicin-loaded platelets to accumulate at the tumor site. In addition, a reduced accumulation of doxorubicin-loaded platelets in the heart tissue was also observed, highlighting that via this approach, cardiotoxic side effects associated with doxorubicin can be minimized [90]. Furthermore, platelets’ surface can be functionalized with various moieties that could actively target tumor cell receptors. An antibody against programmed death ligand-1 (PD-L1) is gaining a lot of attention to selectively capture residual circulatory tumor cells (CTCs) [91]. Platelets coated with anti-PD-L1 were engineered by Gu and his colleagues to reduce, after resection, tumor recurrence in triple-negative breast carcinoma and melanoma in a mouse model and demonstrated that they successfully inhibited tumor recurrence when compared to controls [92]. In another study by Hu et al. [93], platelets bound to anti-PD-L1 and conjugated to hematopoietic stem cells were intravenously administered in a C1498 leukemia mouse model to target tumor cells. The conjugated system migrated to the bone marrow and released anti-PD-L1, thus increasing T-cell migration and cytokine production, resulting in increased survival. This is extremely interesting as platelets could be further employed to bispecifically target the aggregation process along with the inclusion of a chemotherapeutic agent. Coating nanoparticles with platelet membranes is also an attractive option as these nanoparticles can actively target the site of platelet action [94]. According to in vivo experiments performed on mice, platelet membrane-coated polymeric nanoparticles loaded with the anti-MM agent bortezomib and tissue plasminogen activator were able to selectively target MM in the bone marrow [95]. The design of ”human nanoplatelets” with a tunable surface and adequate loading capacity by Dai et al. represents a novel approach for the selective targeting and imaging of early-stage tumors, as they have shown in RPMI8226-derived MM xenotransplants in NOD/SCID mice models [96].

More recently, platelets have been shown to take up bioactive factors secreted by tumor cells, including mRNA from CTCs. The genetic makeup of platelets can potentially be altered by CTCs. Such platelets are called tumor-educated platelets (TEPs) [97]. TEPs could therefore be exploited as a cutting-edge biomarker for early disease detection and intervention. While TEPs have been employed as biomarkers for the early diagnosis of primary thyroid cancer, glioblastoma, ovarian cancer, and renal cell carcinoma [98,99,100,101], their role in hematological malignancies and MM specifically has not been explored.

## 6. Conclusions

The role of platelets in MM has not been extensively investigated. Several multiple myeloma cell lines, including MM.1S, U266, KMS-11, OPM-2, and H929, have been shown to induce platelet aggregation. Additionally, platelets and platelet releasate have been shown to stimulate tumor cell proliferation in vitro, increase tumor engraftment, and decrease mice survival. Regarding platelet activation, although additional research is required, studies in patients have demonstrated increased platelet activation in the premalignant MGUS stage and a lowered activation in the active disease stage. Hemostatic disturbances are frequently observed in newly diagnosed MM patients, and they are also associated with disease treatment. Given the role that platelets play in tumor progression and metastasis, the use of platelets as vehicles has gained popularity in recent years. ”human nanoplatelets” and ”platelet membrane coated nanoparticles” with a tunable surface area are explored for diagnosis and to target tumor cells. Such approaches would be of great potential for the early detection of MM, including the premalignant stages, as well as in the disease management.

## Figures and Tables

**Figure 1 ijms-24-15855-f001:**
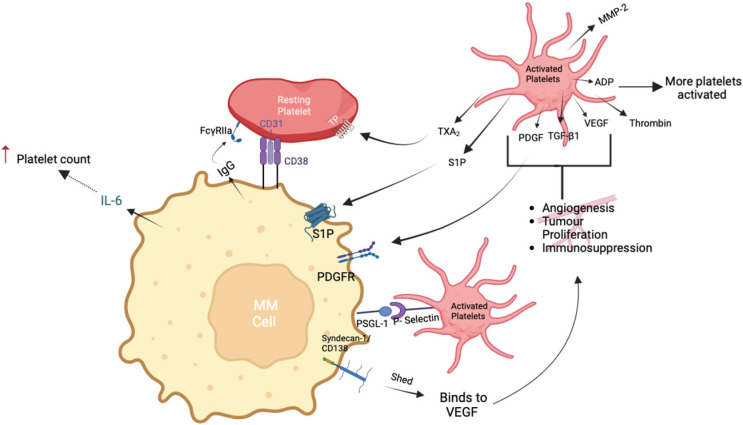
Interaction between MM cells and platelets. MM cells express: CD38, S1P, PDGFR, PSGL-1, and Syndecan-1. CD38 could bind to CD31 expressed on resting platelets, whereas PSGL-1 could bind to P-selectin expressed on activated platelets. MM cells release IL-6 and several immunoglobulins, including IgG. IL-6 can indirectly increase platelet count by increasing thrombopoietin production. IgG binds low-affinity Fc receptor (FcR) for the constant region of IgG (FcγRIIa). During activation and subsequent aggregation, platelets release several factors including TGF-β1, PDGF, S1P, and VEGF. PDGF and S1P have affinity for PDGF and S1P receptors, respectively, resulting in angiogenesis and tumor proliferation. Syndecan-1 shed from MM cells can also bind to VEGF, promoting angiogenesis. PSGL-1: P-selectin glycoprotein ligand-1; ADP: adenosine diphosphate; IgG: immunoglobulin G; IL-6: interleukin-6; PAR: protease-activated receptor; TGF-β1: transforming growth factor β1; PDGF: platelet-derived growth factor; PDGFR: platelet-derived growth factor receptor.

**Table 1 ijms-24-15855-t001:** Revised International Staging System (R-ISS).

R-ISS Stage	Criteria	
I	Sβ2M ^1^	<3.5 mg/L
	Serum albumin	≥3.5 g/dL
	Chromosomal abnormalities by iFISH ^2^	Standard risk
	LDH ^3^	Normal (<upper normal limit)
II	Not R-ISS stage I or III	
III	Sβ2M **and either** Chromosomal abnormalities by FISH **or** LDH	≥5.5 mg/L *High risk High (>upper normal limit)

^1^ Sβ2M—serum β2 microglobulin; ^2^ FISH—fluorescence in situ hybridization; ^3^ LDH—lactate dehydrogenase. Taken from [10]. *High risk is defined as presence of del(17p) and/or translocation t(4;14) and/or translocation t(14;1.6)

## Data Availability

Not applicable.

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
