# Peer review of "Bleeding and Thrombosis in Multiple Myeloma: Platelets as Key Players during Cell Interactions and Potential Use as Drug Delivery Systems"

_ijms, 2023, doi:10.3390/ijms242115855_

Round 1

Reviewer 1 Report (New Reviewer)

Comments and Suggestions for Authors

Overall this is a very interesting topic however the approach of the topic is not ideal and it requires a major re-write. 

- The introduction is not very relevant as a lot of reference is made to the diagnosis and management of myeloma and I believe it should focus on thrombosis and hemorrhagic complications of patients in the MGUS, SMM and MM spectrum. The introduction should focus on these areas specifically rather than the diseases generally. 

Lines 103-106:

Thrombocytosis and thrombotic risk in multiple myeloma is not well described and explained, it should be rephrased to be clearer to the reader. 

Also there is no good description of the underlying well described thrombotic and hemorrhagic mechanisms. 

Line 156: please note than monocytes express also TF after activation. 

Please mention and explain which cells in the bone marrow express P-selectin glycoprotein ligand (PSGL-1) since it is mentioned in the bone marrow microenvironment. 

Lines 186-187: Why do patients with platelet hyporeactivity have inferior survival? Is there a mechanism or data to support this? 

Regarding the figure it is not clear whether MM links via Fc gama or immunoglobulins ? 

The authors should also mention the role of megakaryocytes and how the can change the phenotype of platelets based on the bone marrow microenvironment 

Comments on the Quality of English Language

Please ask for a review of the English language as it requires extensive editing. 

Author Response

Dear Reviewer,

Thank you for giving us the opportunity to revise and re-submit our manuscript titled “Bleeding and thrombosis in multiple myeloma: platelets as key players during cell interactions and potential use as drug delivery systems”. We appreciate the time and efforts you have taken in giving your valuable feedback.

Here is a point-by-point response to your comments.  

Comment- Overall this is a very interesting topic however the approach of the topic is not ideal and it requires a major re-write. The introduction is not very relevant as a lot of reference is made to the diagnosis and management of myeloma and I believe it should focus on thrombosis and hemorrhagic complications of patients in the MGUS, SMM and MM spectrum. The introduction should focus on these areas specifically rather than the diseases generally. 

Response- We believe that including in the introduction an overview of MM and its clinical management is critical for the understanding of the thrombosis and haemorrhagic complications associated with MM, which is in fact the main focus of this review.

Comment- Lines 103-106: Thrombocytosis and thrombotic risk in multiple myeloma is not well described and explained, it should be rephrased to be clearer to the reader. 

Response- Thank you for this observation. We have now omitted this sentence to avoid confusion, particularly in light of the fact that there are few cases of thrombocytosis in MM patients, and thrombotic risk is not mentioned in those reports.

Comment: Also there is no good description of the underlying well described thrombotic and hemorrhagic mechanisms. 

Response: We have now emphasised in the manuscript that the thrombotic and haemorrhagic mechanisms are not well-defined in MM. However, we have now included description of these mechanisms in solid tumours with potential links to haematological cancers.

Comment Line 156: please note than monocytes express also TF after activation. 

Response: This has now been included in the text along with a reference (Ivanov, I. I.; Apta, B. H. R.; Bonna, A. M.; Harper, M. T. Platelet P-Selectin Triggers Rapid Surface Exposure of Tissue Factor in Monocytes. Scientific Reports 2019 9:1 2019, 9 (1), 1–10. https://doi.org/10.1038/s41598-019-49635-7)  

Comment- Please mention and explain which cells in the bone marrow express P-selectin glycoprotein ligand (PSGL-1) since it is mentioned in the bone marrow microenvironment. 

Response- We have now included that leukocytes and myeloid cells also express PSGL-1.

Comment- Lines 186-187: Why do patients with platelet hyporeactivity have inferior survival? Is there a mechanism or data to support this? 

Response- We now clarify that the authors in the paper 78 (Atalay, F.; AteÅŸoʇlu, E. B.; Yildiz, S.; Firatli-Tuglular, T.; KarakuÅŸ, S.; Bayik, M. Relationship of P-Selectin Glycoprotein Ligand-1 to Prognosis in Patients With Multiple Myeloma. Clin Lymphoma Myeloma Leuk 2015, 15 (3), 164–170. https://doi.org/10.1016/J.CLML.2014.09.005) state that MM patients lacking PSGL-1 have lower survival; however, a potential mechanism was lacking. We proposed that this could be attributed to the platelets’ inability to get activated (i.e. platelet hyporeactivity) and therefore could not bind to the receptors on plasma cells [61].  We have now removed this statement as the mechanisms underlying platelet hyporeactivity in MM are under-defined and further work is required to fully elucidate this.

Comment- Regarding the figure it is not clear whether MM links via Fc gama or immunoglobulins? 

Response- We have now clarified in the figure legend (lines 205-206) that MM cells release IgG which binds to low-affinity Fc receptor (FcR) for the constant region of IgG (FcγRIIa).

Comment- The authors should also mention the role of megakaryocytes and how the can change the phenotype of platelets based on the bone marrow microenvironment 

Response- This is a very interesting topic to explore. In our extensive literature search we found a very limited number of studies on how the bone marrow microenvironment alters the platelet phenotype. Winkelmann et al found that the megakaryocyte ploidy in post-mortem bone marrow samples of patients with metastatic carcinoma was significantly higher than in controls which the authors speculated would explain platelet heterogeneity present in the disease. Nevertheless, none of these studies were specific to MM, hence we believe that including these studies is beyond the scope if this review.

Comment- Please ask for a review of the English language as it requires extensive editing. 

Response- We have edited our manuscript by correcting typographical and grammatical errors.

Reviewer 2 Report (New Reviewer)

Comments and Suggestions for Authors

The article entitled “Bleeding and thrombosis in multiple myeloma: platelets as key payers during cell interactions and potential use as drug delivery systems” reviews the biological aspects of the hemostatic dysregulation in Multiple Myeloma with particular attention on platelet function. This topic is particularly interest as the hemostatic complications heavily impact on the outcome of the Multiple Myeloma. The structure is well-organized and the concepts are clear and exhaustive. I have not any observations to make. Therefore, I think that this article is suitable for publication in its current version.

Author Response

Dear Reviewer,

Thank you very much for your comments and for the time you have invested reviewing our manuscript titled “Bleeding and thrombosis in multiple myeloma: platelets as key players during cell interactions and potential use as drug delivery systems”. It is greatly appreciated.

Round 2

Reviewer 1 Report (New Reviewer)

Comments and Suggestions for Authors

I am happy with the revisions and have no further comments. 

This manuscript is a resubmission of an earlier submission. The following is a list of the peer review reports and author responses from that submission.

Round 1

Reviewer 1 Report

Comments and Suggestions for Authors

The topic raised by the authors is unique and very interesting, but contrary to the reader's expectation of acquiring comprehensive knowledge on this topic "the role of platelets in multiple myeloma", this review described mainly the microenviroment factors in multiple myeloma and the role of platelets in solid tumors. 

I understand the situation. As the authors said, the role of platelets in MM has not been extensively investigated, so it is diffucult to make a review article on this particular topic based on the few evidence.

So, I think the author should change the title and the abstract, the structure of manuscript. Otherwise readers might be dissapointed. 

Minor points

Typo in Table 1 t(14;1.6) should be t(14;16).

Line 56: Deletion of chromosome 13q is not  included in high risk cytogenetics. Deletion of chromosome 17p is included. 

Reviewer 2 Report

Comments and Suggestions for Authors

The title of this manuscript is ” the role of platelets in MM". There are little evidence for the relationships between platelets and myeloma cells, and the theme is interesting for clinicians and researchers. However, this manuscript have some problems as indicated below. 

#1  Most part of the first and second chapter, "Introduction" and "BM microenvironment and MM" is not relevant to the main theme, "platelets and MM". Inappropriate words such as B plasma cell, DDK-1 are frequently seen. 

#2  The relationships between platelets and MM is only described in chapter 4. Even in chapter 4, only a few references match the theme, and more than half of the reference in this chapter belong to rare case reports, which is not applicable for most patients with myeloma in the real-world.  

Comments on the Quality of English Language

There are some odd wordings especially for the use of medical words. I suggest to have some external language editing done by a specialist of this field.